evolution, behaviour

*Heliconius*, chemical signatures, mate choice, speciation, reproductive isolation

**Author for correspondence:**
C. Salazar
e-mail: camilo.salazar@urosario.edu.co

# Chemical signals act as the main reproductive barrier between sister and mimetic *Heliconius* butterflies

M. F. González-Rojas[1], K. Darragh[2], J. Robles[3], M. Linares[1], S. Schulz[4], W. O. McMillan[5], C. D. Jiggins[2], C. Pardo-Diaz[1] and C. Salazar[1]

[1]Department of Biology, Faculty of Natural Sciences and Mathematics, Universidad del Rosario, Bogota 111221, Colombia
[2]Department of Zoology, University of Cambridge, Cambridge, Cambridgeshire CB2 3EJ, UK
[3]Department of Chemistry, Pontificia Universidad Javeriana, Bogota, Colombia
[4]Institute of Organic Chemistry, Technische Universität Braunschweig, Braunschweig, Germany
[5]Smithsonian Tropical Research Institute, Panama

CS, 0000-0001-9217-6588

Colour pattern is the main trait that drives mate recognition between *Heliconius* species that are phylogenetically close. However, when this cue is compromised such as in cases of mimetic, sympatric and closely related species, alternative mating signals must evolve to ensure reproductive isolation and species integrity. The closely related species *Heliconius melpomene malleti* and *H. timareta florencia* occur in the same geographical region, and despite being co-mimics, they display strong reproductive isolation. In order to test which cues differ between species, and potentially contribute to reproductive isolation, we quantified differences in the wing phenotype and the male chemical profile. As expected, the wing colour pattern was indistinguishable between the two species, while the chemical profile of the androconial and genital males' extracts showed marked differences. We then conducted behavioural experiments to study the importance of these signals in mate recognition by females. In agreement with our previous results, we found that chemical blends and not wing colour pattern drive the preference of females for conspecific males. Also, experiments with hybrid males and females suggested an important genetic component for both chemical production and preference. Altogether, these results suggest that chemicals are the major reproductive barrier opposing gene flow between these two sister and co-mimic species.

## 1. Introduction

The mechanisms by which species maintain their integrity are diverse and involve a combination of multiple signals of intra- and interspecific communications such as chemical, visual, auditory and tactile cues [1–5]. In particular, sexual communication in insects involves long- and short-range pheromones, which play multiple roles [3,4,6,7]. For instance, pheromones can communicate the mating status of females [8,9], quality and age of males [10,11], quality and quantity of nuptial gifts [12], body size [13], dominance status [14] and degree of relatedness [15]. Also, chemicals play an important role in mate choice and species recognition [16,17]. In particular, pheromones mediate mate choice in many insects including flies (*Drosophila*), grasshoppers (*Chorthippus parallelus*), leaf beetles (*Chrisochus*) and stick insects (*Timema*) [16–23].

In Lepidoptera, males and females produce volatile and non-volatile compounds, suggesting that chemical communication plays a critical role in inter- and intraspecific communications [24–26]. In *Bicyclus anynana*, visual and chemical cues are equally important for mate choice, and females recognize

heterospecific males based on their pheromones [27]. Also, males of *Heliconius charithonia*, which engage in pupal mating by copulating with females as they eclose, can identify the sex of a conspecific pupa based on sex-specific compounds [28,29]. Moreover, chemical cues seem to mediate species recognition among mimetic and distantly related *Heliconius* species where conflicts between mimicry and sexual communication may arise [30–32]. This is supported by the fact that males of *Heliconius* have species-specific mixtures in their wing androconia (i.e. specialized male wing scales that produce scents) [32,33].

In *Heliconius*, mate discrimination between closely related species relies on the wing colour pattern [34,35]. For example, the sister species *H. melpomene* and *H. cydno* (divergence approx. 1.5–2 Ma) [36], which are sympatric across Central America and the Andes, not only differ in habitat use but also in wing coloration [37–39]. In fact, multiple experiments show that males prefer to court females exhibiting their own colour pattern [34,40,41]. By contrast, the phylogenetically close *H. melpomene malleti* and *H. timareta florencia* mimic each other and coexist in sympatry in the Andes in southeastern Colombia (electronic supplementary material, figure S1). Despite their phenotypic resemblance, this species pair shows strong premating ecological isolation (differences in host plant preference) as well as strong reproductive isolation tested in no-choice experiments [40,42]; even so, a low number of hybrids are found in nature (approx. 2%) [31,40]. Therefore, the strong reproductive isolation between *H. m. malleti* and *H. t. florencia* implies that sexual isolation is mediated by cues other than the colour pattern, such as chemical cues [31,32,41].

In agreement with this hypothesis, previous studies showed that the two species differ in their androconia and genital chemical composition, although these studies included few individuals [30,32]. Furthermore, we previously showed that females of *H. m. malleti* and *H. t. florencia* strongly discriminated against conspecific males which have their androconia experimentally blocked, affecting reproductive success with implications for reproductive isolation. This suggests that chemical signalling is important in mate choice in *Heliconius* butterflies [43].

Nonetheless, we need to investigate more on (i) the preference of females for conspecific versus heterospecific male chemical blends in order to understand their role and importance in reproductive isolation, and (ii) the inheritance patterns of both male chemical production and female preference for them, and thus contribute to our understanding of the genetic architecture of speciation. Here, we used a combination of behavioural and chemical analyses to get a better understanding of reproductive isolation meditated by chemical signals in *Heliconius* butterflies.

## 2. Material and methods

### (a) Quantification of the wing phenotype

To quantify colour, we used wings of wild males of *H. t. florencia* and *H. m. malleti* deposited in the Colección de Artrópodos de la Universidad del Rosario (CAUR229) (electronic supplementary material, table S1) and evaluated whether *H. m. malleti* and *H. t. florencia* exhibit differences in the wing phenotype. In order to do this, we scanned ventral and dorsal forewings and hindwings of 43 *H. m. malleti* and 45 *H. t. florencia* using a high-resolution Epson Perfection V550 flatbed scanner, in RGB colour format with a resolution of 2400 dpi. Right-side wings were always used. Then, we used ImageJ [44] to place a set of 34 landmark coordinates

per individual (dorsally and ventrally; electronic supplementary material, figure S2A). These landmark coordinates were analysed in the R package Patternize [45] to quantify variation in wing colour patterns (band size and shape). This package extracts, transforms and superimposes colour patterns to finely quantify the variation of colour pattern among species, and performs a principal component analysis (PCA) on the binary representation of the aligned colour pattern obtained from each sample with the *sumRaster* function [45]. We then tested differences in the wing pattern among species using a multivariate analysis of variance (MANOVA) in R based on a subset of PCs (those that explain greater than 95% of the variation).

In addition, we used tpsDig2 [46] to place 32 landmark coordinates on the outline of both forewing and hindwing (dorsally; electronic supplementary material, figure S2B). These landmark coordinates were superimposed using a general Procrustes analysis (GPA) in the R package 'geomorph' [47–49]. The resulting coordinates in the tangent space were used as shape data, while the log-transformed centroid size [48] was used as a size estimate [50]. Differences in wing size among species were investigated with a one-way ANOVA with size as a dependent variable and species as a factor, followed by Tukey's pairwise comparison test. Differences in wing shape among species were tested using a Procrustes MANOVA applied to the aligned landmark configurations. This was done using the *procD.lm* function in the 'geomorph' R package [47].

### (b) Wild sampling and interspecific crosses

We collected wild individuals of *H. t. florencia* and *H. m. malleti* in the localities of Sucre and Doraditas (Colombia) that were taken to the insectaries of the Universidad del Rosario in La Vega (Colombia) to establish stock populations for the behavioural experiments and chemical analyses. Larvae were reared on *Passiflora oerstedii*, while adults were provided with *Psiguria* sp. as a pollen source and supplied with approximately 20% sugar solution.

We also used the stock populations to perform interspecific crosses between *H. t. florencia* and *H. m. malleti.* To do so, a female of *H. t. florencia* was mated with a male of *H. m. malleti* (the reciprocal cross was successful three times, but the female always died before laying eggs). Then, two $F_1$ males were backcrossed to pure *H. t. florencia* females (crosses towards females of *H. m. malleti* consistently failed). In all cases, eggs were collected daily and placed in small plastic pots. Larvae were reared individually to avoid cannibalism, and right before pupation, they were transferred to bigger plastic pots until eclosion. The two backcrosses towards *H. t. florencia* produced 25 males and 25 females: all males were processed to characterize the composition of their chemical blends, while 24 females were used in behavioural experiments testing for female preference (see below).

Field collections and insectary rearing were conducted under permit no. 530 issued by the Autoridad Nacional de Licencias Ambientales of Colombia (ANLA). Rearing conditions and experimental crosses were approved by the Ethics Committee of Universidad del Rosario (approval no. CEI-ABN026-000155).

### (c) Behavioural experiments

To test female preference for colour pattern and chemical cues of conspecific males, we conducted two types of behavioural experiments in triads: (i) altering the wing phenotype of males and (ii) perfuming males with the heterospecific chemical blends. All experiments were conducted from 7.00 to 13.00, checking every 30 min for mating; the experiments stopped when mating occurred. For each experiment, mature males (at least 10 days old) were randomly selected from the stock population, while females were used as soon as they became available. If no mating occurred on the first day, we repeated the experiment the next day using the same butterflies; by contrast, mated males or

females were never re-used. If no mating occurred after the second day, the experiment stopped. For the first hour of each experiment, we observed female behaviour towards the males. These behaviours were recorded only when a male was actively courting the female. Observations were divided into 1 min intervals, and we recorded female behaviours of 'acceptance' or 'rejection' previously defined [43,51]. Specifically, we recorded the following acceptance behaviours: flutter (slow and moderate wing flapping), fly towards (flight facing towards the male), slow flat (slow rhythmic flight), and wings open and exposed (wings open and abdomen relaxed). Similarly, we recorded the following rejection behaviours: fly away (flight away from the male), tucked up (alighted with wings closed and abdomen concealed within the wings), rapid and erratic flutter (high frequency flutter of her wings and abdomen raised when the male is in close proximity), and wings opened and abdomen bend (wings opened and abdomen raised, but without wing fluttering).

In the triads that tested female preference for colour pattern, a single 1-day-old virgin female was presented with two conspecific males of at least 10 days old. One of the males (treatment) had his forewing and hindwing completely blacked out using a black marker (COPIC 100), thus hiding his wing colour pattern. The second male (control) had his forewing and hindwing painted with a colourless marker (COPIC 0); in this way, the male kept his phenotype unaltered, but we controlled for any odour effect of the marker. We tested a total of 20 females per species.

For the triads that tested female preference for chemical signals, we first prepared extracts from sexually mature males of each species by dissecting and mixing the androconia region of five conspecific individuals of the same age and soaking them in 200 µl of hexane for 1 h. After this incubation, the solvent was transferred to a new vial and stored at −20°C until needed. Then, a single 1-day-old virgin female was presented with two sexually mature conspecific males. Both males were initially treated with transparent nail varnish applied on the dorsal side of their hindwing in order to block their androconia and thus block their natural emission of chemicals. Then, one of the males (control) was perfumed by spreading the conspecific hexane extract in the androconia region, whereas the second male (treatment) was perfumed by spreading the heterospecific hexane extract. A total of 19 females of *H. t. florencia* and 18 females of *H. m. malleti* were tested. In order to investigate how long the hexane extract remains in the wings of the perfumed males before completely evaporating (which can potentially affect our results), we blocked the hindwing androconia of nine males of each species using transparent nail varnish and then, we re-perfumed this region by spreading the heterospecific hexane extract. We left these males fly in an insectary and dissected their androconia at 1, 30 or 60 min, and soaked this tissue in 200 µl of ultrapure dichloromethane (Merck UniSolv) to be later analysed by gas chromatography/mass spectrometry (GC/MS; see below).

We also studied hybrid female preference. A single virgin hybrid female had to choose between two males, one *H. t. florencia* and one *H. m. malleti*. A total of 18 $F_1$ and 24 backcross 1-day-old virgin females were tested. These triads were conducted in the same way as the previous behavioural experiments. Female acceptance or rejection behaviours were also recorded.

The mating outcome was analysed with a binomial test. We also used a generalized linear mixed model (GLMM) with a binomial error distribution and a logit link function to test if females responded differently to control and treatment males. The response variable was derived from those minutes where at least one of the males courted the female regardless of her response (either 'acceptance' or 'rejection'). Significance was determined by using likelihood ratio tests comparing models with and without the male type included as an explanatory variable. In order to avoid pseudoreplication, individual female was included as a random effect in all models. All statistical analyses were performed with R v. 3.3.2 [52], using the packages lme4 [53], ggplot2 [54], car [55] and binom [56] following Darragh *et al.* [43].

## (d) Characterization of chemical profiles

To determine the chemical composition of the volatile compounds of the androconia and genitalia in males of *H. m. malleti*, *H. t. florencia* and their hybrids ($F_1$ and backcrosses), we dissected both tissues from adults and placed them individually in 200 µl of ultrapure dichloromethane (Merck UniSolv) in 2 ml glass vials and soaked for 1 h. After this incubation, the solvent was transferred to a new vial and stored at −20°C. We used dichloromethane since it is sufficiently volatile for extracts to be concentrated without exposing them to high temperatures, it is non-flammable, and penetrates wing scales better than hexane leading to higher extract titres [57,58].

These extracts were analysed by GC/MS at the Smithsonian Tropical Research Institute following a previous protocol [32]. Prior to the GC/MS, samples were evaporated under ambient air at room temperature. Then, we quantified the compounds found in the extracts using a Hewlett-Packard GC model 5977 mass-selective detector, connected to a Hewlett-Packard GC model 7890B and equipped with a Hewlett-Packard ALS 7693 autosampler. A BPX-5 fused silica capillary column (SGE, 25 m × 0.22 mm, 0.25 µm) was used. Injection was performed in a splitless mode (250°C injector temperature) with helium as the carrier gas with a constant flow of 1.2 ml min$^{-1}$. The temperature programme started at 50°C, held for 5 min and then rose to 320°C with a heating rate of 5°C min$^{-1}$. We used 2-tetradecyl acetate (200 ng) as an internal standard. Components were identified by the comparison of mass spectra and gas chromatographic retention index with reference samples from the Schulz lab collection (Institute of Organic Chemistry, Technische Universität Braunschweig). Relative concentrations were determined by peak area analysis by GC/MS.

To evaluate species differences in compound composition, we implemented a dimension reduction (PCAs) using the software PAST v. 3.0 [59]. We retained the components accounting for 95% of the variance, and those were used to conduct a discriminant analysis. A MANOVA was performed in candisc R package [60]. Finally, using the means of the relative concentrations of each compound, we established the relationship among *H. t. florencia*, *H. m. malleti*, $F_1$ and backcross males for androconia and genital bouquet (i.e. mixture of volatiles), and calculated dendrograms using the Euclidian distance. Both dendrogram and compound composition were visualized using the function heatmap in R (v. 3.5.0).

# 3. Results

## (a) Quantification of the wing phenotype

We found that *H. m. malleti* and *H. t. florencia* are significantly different in wing size, both in the forewing and hindwing (ANOVA $F_{1,82} = 63.894$, $p < 0.01$ and $F_{1,82} = 82.587$, $p < 0.01$, respectively, electronic supplementary material, figure S3A, B). Forewings and hindwings of *H. t. florencia* are consistently larger than those of *H. m. malleti*. In terms of shape, the forewing is statistically different between species (MANOVA, $F_{1,78} = 28.09$, $p < 0.01$), but not the hindwing (MANOVA, $F_{1,78} = 2.969$, $p > 0.01$; $\alpha = 0.01$; electronic supplementary material, figures S3C,D and S4A). In the forewing, the most variable landmark coordinates were landmark 3, landmark 4 (located in the distal part of the costal margin, near the wing apex; electronic supplementary material, figures S2C,D and S4B) and landmark 15 (located in the middle of the inner margin; electronic supplementary material, figures S2E,F and

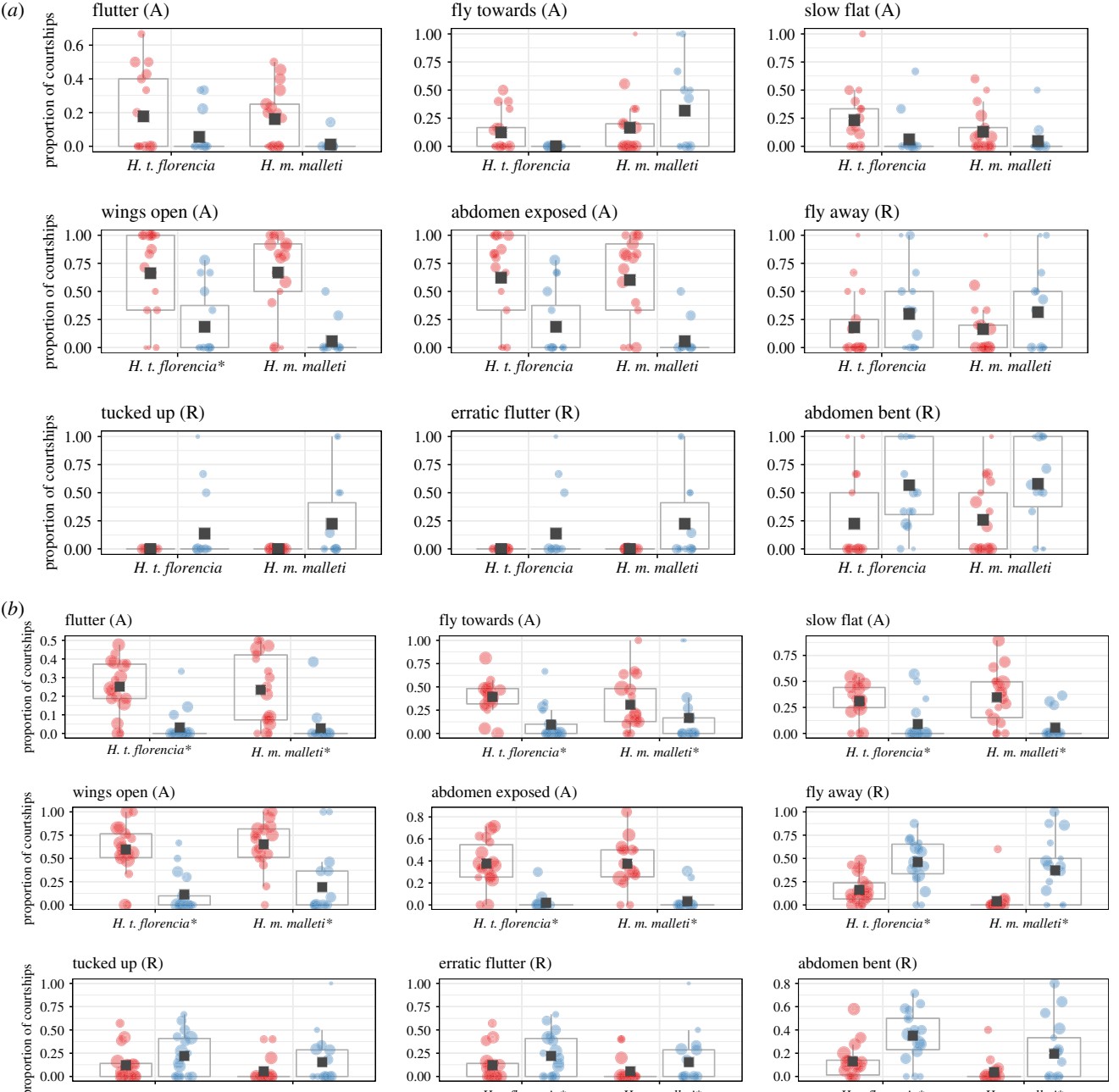

**Figure 1.** (*a*) Proportion of courtships that resulted in female behavioural responses towards control and treatment males. (*b*) Female behavioural responses towards conspecific males 'perfumed' with a hexane extract from five males of either *H. m. malleti* or *H. t. florencia*. Behaviours are classified as acceptance (A) or rejection (R). Control males are represented in red (left) and treatment males in blue (right). Means are marked with a black square and boxplots mark the interquartile ranges. Asterisk next to the species's name is indicative of statistically significance ($[\alpha] = 0.01$) according to the GLMM. (Online version in colour.)

S4C). Thus, the significant shape variation between the two species appears to be linked with the length of the forewing (longer in *H. t. florencia*) and the curvature of the inner margin (deeper in *H. t. florencia*).

The colour pattern comparison between the two species suggested subtle differences in the wing colour pattern (electronic supplementary material, figure S5). The shape (PC2) of the three wing elements investigated (forewing band, forewing 'dennis' patch and hindwing rays) did not differ between species, but their size (PC1) was slightly different (electronic supplementary material, figure S5). Despite this, none of these differences were statistically significant, indicating that the two species are almost indistinguishable in terms of wing colour pattern.

### (b) Behavioural experiments

Altering the wing pattern of males had no effect on mating probability. In all 40 experiments, both *H. t. florencia* and *H. m. malleti* females mated readily with conspecific males completely blacked (exact binomial test, $p = 0.55$ in both cases; electronic supplementary material, figure S6). Consistent with this finding, we found no differences in the female response towards control and treatment males in all but one of the acceptance and rejection behaviours we assayed (figure 1*a*; electronic supplementary material, table S2). The exception was the observation that female *H. t. florencia* kept their wings open for a larger proportion of the time during courtship by control males relative to male whose wing patterns were eliminated (open wings; figure 1*a*).

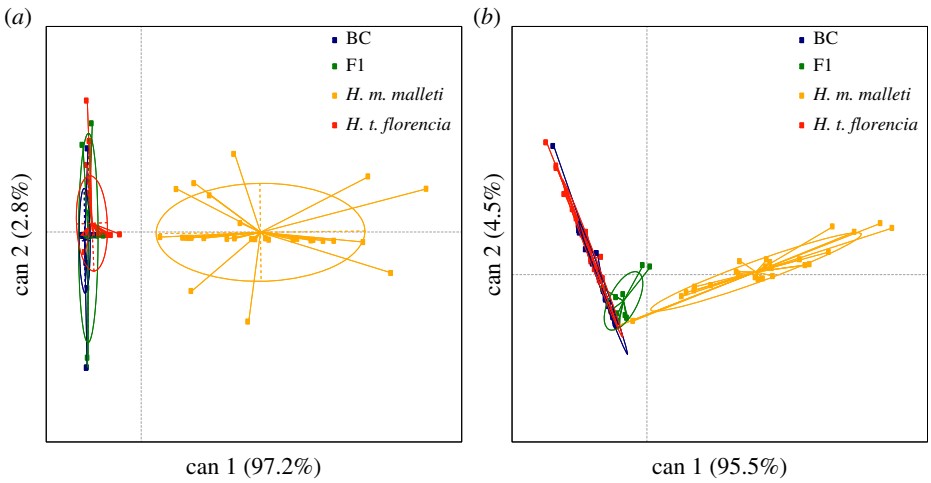

**Figure 2.** (*a*) Discriminant analysis based on the individual composition of wing androconia extracts of males of *H. m. malleti*, *H. t. florencia*, F₁ and backcrosses (BCs). (*b*) Discriminant analysis based on the individual composition of abdominal gland extracts of males of *H. m. malleti*, *H. t. florencia*, F₁ and BCs. (Online version in colour.)

The triads testing female preference for perfumed males ($n = 37$) did not yield any matings. The failure to mate might reflect the rapid evaporation rate of the chemical extracts. For example, the concentration of octadecanal decreased 25% in the first 30 min and 82% after 60 min (electronic supplementary material, figure S7 and table S4). Similarly, the concentration of syringaldehyde decreased 25% in the first 30 min and 47% after the first hour. Nevertheless, we observed significant differences in female behavioural responses towards treatment and control males (figure 1*b*; electronic supplementary material, table S3). Specifically, in all behaviours tested, females exhibited acceptance behaviours towards males perfumed with the hexane extract of their own species and consistently rejected males perfumed with that of the other species.

F₁ and backcross females obtained from mating an F₁ female to a *H. t. florencia* male were reluctant to mate either parental species. In these trials, F₁ mated at 33% frequency and backcross females at 37% frequency; in all cases matings were with *H. t. florencia* males (electronic supplementary material, figure S8). This agrees with previously unpublished no-choice experiments where F₁ mated at 25% frequency and backcross females at 37% frequency, always with *H. t. florencia* males (electronic supplementary material, table S8). Consistently, hybrid females were more likely to perform acceptance behaviours towards *H. t. florencia* males, while rejection behaviours were observed more often towards *H. m. malleti* males in most of the traits measured (electronic supplementary material, table S5).

### (c) Characterization of chemical profiles

We analysed 100 wing androconia and 95 abdominal gland extracts from males of *H. m. malleti* ($n_{and} = 31$, $n_{gland} = 28$), *H. t. florencia* ($n_{and} = 33$, $n_{gland} = 32$), F₁ ($n_{and} = 11$, $n_{gland} = 11$) and backcrosses to *H. t. florencia* ($n_{and} = 25$, $n_{gland} = 24$). Males of both species presented a common composition in the wing androconia extract, and the most notable differences were in terms of the concentration of individual compounds in the blend (electronic supplementary material, table S6). The compounds of the androconial region in *H. m. malleti* were mainly alkanes (28%), aldehydes (16.27%) and unknown compounds (18%), with octadecanal being the most abundant compound (1094.91 ng in *H. m. malleti* compared with 1.40 ng in *H. t. florencia*). In *H. t. florencia*, the androconial bouquet was

composed of alkanes (37%) and esters (14%), with syringaldehyde and heneicosane being the most abundant compounds (electronic supplementary material, figure S9). The androconial composition of F₁ and backcross males was very similar to that of *H. t. florencia* but showed higher individual variation (electronic supplementary material, figures S10, S11 and table S6). Consistently, the discriminant analysis revealed a discrete group formed by *H. t. florencia*, F₁ and backcross males, while *H. m. malleti* formed an independent cluster (figure 2*a*; MANOVA, $F_{1,3} = 13.347$, $p < 0.01$). A post hoc Tukey's test showed that F₁ and backcross males differed significantly from *H. m. malleti* males ($p < 0.01$; in both cases) but not from those of *H. t. florencia* ($p > 0.01$; in both cases).

The abdominal gland bouquet of males was chemically more diverse than that of the androconia (electronic supplementary material, table S7). Alkanes, esters and lactones were the major compounds present in the abdominal gland bouquet of *H. m. malleti* and *H. t. florencia*. β-ocimene and heneicosane largely dominated the abdominal gland bouquet of *H. m. malleti* males, while that of *H. t. florencia* was mainly composed of ethyl oleate, butyl oleate, isopropyl oleate and (Z)-9-octadecen-13-olide (electronic supplementary material, table S7 and figure S12). As in the androconia, the abdominal gland bouquet of F₁ and backcross males was more similar to *H. t. florencia*, although some individuals had β-ocimene (electronic supplementary material, figures S13 and S14). The discriminant analysis revealed a discrete group composed of *H. t. florencia*, F₁ and backcross males which differentiate from a cluster formed only by *H. m. malleti* (figure 2*b*; MANOVA, $F_{1,3} = 12.528$, $p < 0.01$). A post hoc Tukey's test showed that F₁ and backcross males differed significantly from *H. m. malleti* males ($p < 0.01$; in both cases) but not from *H. t. florencia* males ($p > 0.01$, in both cases). Interestingly, both species and their hybrids showed putative defensive secretions in their abdominal gland, namely 2-sec-butyl-3-methoxypyrazine in the abdominal gland extracts of *H. t. florencia*, F₁ and backcrosses and 2-isobutyl-3-methoxypyrazine in the abdominal gland extracts of *H. m. malleti*, F₁ and backcrosses (electronic supplementary material, table S7 and figure S13). These specific compounds are known to deter predators in the wood tiger moth [61], and in general, methoxypyrazines are compounds frequently found in the chemical defences of aposematic insects [62–64].

## 4. Discussion

Variations in sexual cues and preferences have major implications for speciation, as they can cause reproductive isolation. The role of visual cues and visual preference in triggering reproductive isolation has been well documented in animal taxa that have diverged in the presence of gene flow [65–67]. By contrast, while it is widely acknowledged that chemical signals play a role in animal premating isolation [68,69], their role as drivers of sympatric speciation is much less studied. Lepidoptera is the order that has the most information on volatiles involved in sexual recognition: more than 2000 sex pheromones have been identified in moths, and an additional eight have been identified in day flying butterflies [70–72]. The function of sex chemicals in promoting species recognition and ensuring reproductive isolation in moths has been extensively demonstrated [72], while in butterflies, experimental evidence of such phenomena is less abundant [10,71,73,74]. This is probably because researchers usually assume that butterflies should mainly rely on mating systems based on visual signals suited to their diurnal habits.

We found that *H. t. florencia* and *H. m. malleti* are nearly identical in terms of wing patterning, and altering this trait with black markers had little effect on the preference of females for mates. This accords with our earlier observations where experiments with wing models washed in hexane revealed that males of *H. t. florencia* approached and courted models of *H. m. malleti* as much as theirs [42]. These results suggest that the wing phenotype is not the cue that maintains species integrity between these mimetic pair. Instead, strong mating isolation appears to be largely driven by chemical signals. Our previous work demonstrated that females of *H. m. malleti* and *H. t. florencia* strongly discriminated against conspecific males that have their wing androconia experimentally blocked [43]. Consistently, we found that females showed more acceptance behaviour and less rejection behaviour towards males perfumed with a conspecific extract relative to those perfumed with the heterospecific extract. Intriguingly, no matings were observed in these perfuming experiments, which could be partly due to the rapid evaporation of the androconial perfume applied. However, since these experiments did not alter the chemicals in the abdominal gland, the lack of mating also indicates that the sole presence of the abdominal scent is not enough to ensure successful mating, and the androconial scent is always needed. Even more, the compound composition in each of these scents possibly needs to occur in specific mixture ratios. This would agree with recent findings in the butterfly *Pieris napi*, where the synergistic processing of two wing male volatile components in a 1 : 1 ratio is necessary for female acceptance [74]. Also, in *Drosophila*, the disruption natural ratios in male cuticular hydrocarbons led to an aversion response in females [75], further suggesting that normal mixture ratios in the chemicals underlying mate acceptance and preference are required. Therefore, it is likely that the abdominal and androconial extracts, and their specific composition, play different roles at different stages of courtship (although this remains to be tested).

We also found that the composition of the male chemical bouquet of *H. m. malleti* and *H. t. florencia* is different, confirming the results previously reported in other studies that included a much lower sample size [30]. The chemical signature of the two species is unique, not only in the androconia but also in the genitalia. This contrasts with the general pattern in Lepidoptera, where closely related species usually display similar chemical signatures, especially in pheromones [70]. In particular, we identified octadecanal and β-ocimene as the main compounds in the androconia and genitalia bouquet of *H. m. malleti*, respectively. Even though octadecanal was also present in *H. t. florencia*, its abundance was much lower in this species. This agrees with recent findings, where octadecanal was found to be abundant in males of *H. m. rosina* and almost absent in those of *H. c. chioneus* [71]. Interestingly, octadecanal is electro-physiologically active in *Heliconius* [71], and β-ocimene is a known anti-aphrodisiac in the genus [76]. Also, the most abundant compounds in the male androconia of *H. t. florencia* were syringaldehyde and heneicosane, known to act as long-range attraction molecules in multiple insect species [77–81]. Similarly, the male bouquet of this species also contained phenylacetaldehyde and limonene, which act as copulating pheromones, hormones and defensive secretions in other Lepidoptera [78]. The marked difference in chemical composition in the genital and wing extracts of males of *H. m. malleti* and *H. t. florencia* may be facilitated by the fact that these species have strong differences in larval host plant use [40], and larval diet is a known factor that affects adult pheromone composition in Lepidoptera [72,82].

Hybrid males ($F_1$ and backcrosses) had fewer and less abundant compounds in their blends compared with the parental species. Interestingly, both $F_1$s and backcrosses had little octadecanal and some hybrid individuals even lacked it at all (just as pure *H. t. florencia* males do), suggesting that the low amount of this compound is heritable and possibly involving few major effect loci. This agrees with findings in *H. m. rosina* and *H. c. chioneus* that suggest a potential monogenic basis and dominant inheritance for octadecanal production [71]. Similarly, hybrid females ($F_1$ and backcrosses) accepted males of *H. t. florencia* more readily than those of *H. m. malleti*, and in some cases, we observed matings with the former species. This again suggests that female preference could be heritable and with a simple genetic basis. In fact, examples in orchid bees and *Drosophila* suggest that the genetics of chemical production and chemical preference have a simple genetics, where the same or few genes are involved [83,84]. However, our data are not enough to draw definitive conclusions on the genetics of the production of sexual chemical cues or female preference. To answer this question, it would be necessary to test the reciprocal $F_1$ (*H. m. malleti* mother × *H. t. florencia* father) and backcrosses towards *H. m. malleti*. However, such crosses are extremely difficult to obtain, perhaps due to intrinsic behavioural sterility as reported for other Lepidoptera [85].

A previous QTL analysis in crosses between *H. melpomene* and *H. cydno* found no genetic linkage among colour pattern and/or colour preference loci with the locus underlying the production of octadecanal [71]. This is unexpected if chemical cues play a major role in reproductive isolation, as theory predicts that traits under divergent selection and those that play a role in premating isolation should be tightly linked in order to facilitate speciation [65,86]. By contrast, in cases where natural selection promotes mimicry convergence due to secondary introgression, thus compromising the visual recognition of conspecific mates, reproductive isolation should weaken, possibly leading to species collapse [31,87]. Yet the total reproductive isolation between *H. m. malleti* and *H. t. florencia* is approximately 97% [31], implying that chemical barriers, along with other barriers such as habitat specialization and host plant use, are sufficient to ensure species integrity. Our hybrid

broods suggest that both sexual chemical production and preference for them probably have a simple genetic basis, and if they happen to be physically linked in the genome, as in the case of visual cues and visual preference [65], then reproductive isolation is easily achieved and maintained.

Overall, this study corroborates the existence of differences between male chemical signatures in a pair of closely related and mimetic species where the wing phenotype is not a recognition trait. These chemical differences are used by females to effectively choose mates, confirming that male sex compounds are necessary to mediate premating reproductive isolation between *H. m. malleti* and *H. t. florencia*. Also, our results suggest that both male chemical production and female preference in *Heliconius* are heritable traits with a simple genetic basis. In fact, recent studies have pinpointed candidate regions implicated in the production of male pheromone components in *Heliconius* [71]. However, to date, the genetic basis controlling female assortative mating behaviours remains largely unknown (but see [88]). This study shows that sexual chemicals are effective cues that contribute to mate recognition in day flying butterflies, and that both sexual chemical production and chemical preference are key components of reproductive isolation that facilitate speciation in the face of gene flow.

Ethics. Field collections and insectary rearing were conducted under permit no. 530 issued by the Autoridad Nacional de Licencias Ambientales of Colombia (ANLA). Rearing conditions and experimental crosses were approved by the Ethics Committee of Universidad del Rosario (approval no. CEI-ABN026-000155).

Authors' contributions. M.F.G.-R. performed the experiment, analysed the data, contributed reagents/materials/analysis tools, wrote the paper, prepared figures and/or tables, and designed the experiments. K.D. contributed reagents/materials/analysis tools and critically revised the manuscript. J.R. prepared figures and/or tables and critically revised the manuscript. M.L. contributed reagents/materials/analysis tools. S.S. contributed reagents/materials/analysis tools, prepared figures and/or tables, and critically revised the manuscript. W.O.M. contributed reagents/materials/analysis tools and critically revised the manuscript. C.D.J conceived and designed the experiments, contributed reagents/materials/analysis tools, and critically revised the manuscript. C.P.-D and C.S. conceived and designed the experiments, contributed reagents/materials/analysis tools, wrote the paper, and critically revised the manuscript.

Competing interests. We declare we have no competing interests.

Funding. This study was funded by COLCIENCIAS (grant no. FP44842-5-2017) and Universidad del Rosario (grant no. QFN-DG001). We thank ANLA-Colombia for granting us the collection permit no. 530. S.S. thanks the Deutsche Forschungsgemeinschaft (DFG; grant no. Schu984/12-1). C.D.J. was funded by the European Research Council (grant no. 339873 Speciation Genetics).

Acknowledgements. We thank Oscar Penagos and Sebastian Sánchez their help rearing butterflies. Steven Van Belleghem helped running Patternize, and Kelsey Bryes gave useful feedback. We also thank the High Performance Computing Service of Universidad del Rosario (CENTAURO).

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
