## [Reviewer comments · Proceedings of the Royal Society B: Biological Sciences]

Review History

RSPB-2019-2646.R0 (Original submission)

Review form: Reviewer 1

Recommendation

Major revision is needed (please make suggestions in comments)

Scientific importance: Is the manuscript an original and important contribution to its field?

Good

General interest: Is the paper of sufficient general interest?

Acceptable

Quality of the paper: Is the overall quality of the paper suitable?

Good

Is the length of the paper justified?

Yes

Should the paper be seen by a specialist statistical reviewer?

No

Do you have any concerns about statistical analyses in this paper? If so, please specify them explicitly in your report.

No

It is a condition of publication that authors make their supporting data, code and materials available - either as supplementary material or hosted in an external repository. Please rate, if applicable, the supporting data on the following criteria.

Is it accessible?

No

Is it clear?

N/A

Is it adequate?

N/A

Do you have any ethical concerns with this paper?

No

Comments to the Author

This manuscript investigated the importance of male wing colour and chemical composition of the androconia for female mating preference, and species discrimination, in two species of *Heliconius*.

It is clear that a lot of work has gone into this manuscript, and it has important implications for the study of species discrimination and sexual selection in mimicry rings, however I have some concerns about interpretation and terminology.

None of the trials where male chemical profile was manipulated resulted in matings, the authors therefore use acceptance and rejection behaviours to conclude that the chemical cues are responsible for female preference. This in and of itself is not an issue, perfuming experiments are tricky after all, but it is never mentioned in the discussion. Given that this is a key part of your study I think it warrants a little more discussion. In particular, as you found differences in genital gland extracts between the species but did not manipulate this in your behavioural experiments, perhaps that played a role? If different extracts play a role at different stages of courtship that may explain why no matings occurred despite females showing some acceptance behaviours .

I am also concerned with the language used to describe the "pheromone" composition results. If my reading is correct you have no way of knowing which components of the extracts are actually being detected and used in mating decisions by these females. Nevertheless they (by necessity) all have equal weight in your statistical analysis. This is a sensible approach to begin characterising the pheromones of these species, but you cannot then call this the pheromone bouquet, or conclude that the pheromone composition of the two species is significantly different - you can only say that the chemical composition of the extracts are different. Some of these chemicals (maybe even all) are pheromones, but you don't yet know which.

Just to be clear- the androconial extract clearly contains pheromones, as blocking them prevented successful mating, but the PCA was performed on chemical composition. This means you have to be very careful when discussing your results as pheromone is the correct term to use for some aspects but not others.

There were also a few things in the methods that were unclear to me. When describing how you tested the longevity of the perfuming method on L200-205 you lost me entirely. Surely by dissecting and extracting the androconia in this way you also risk picking up the males own gland secretions? Can you give a little more information on why you used this method?

On L236-251 it is unclear if you used GC-FID, GC-MS or both.

L353 in the methods you say you used PCA and here you say discriminate analysis. To my knowledge they are similar but not the same. Please clarify.

I also have some more minor comments:

L31 I think you mean “preference of females for conspecific males”

L45 I suggest replacing “inform” with “can communicate”

L70 coexist

L74 “even so” suggests contrasting information. Replace with “and few hybrids etc etc”

L459 “inheritable” is a bit confusing, replace with “heritable”

Figures: I appreciate that you have limited figures but the decision not to have the figure for the behavioural response to chemical manipulation is a bit strange to me.

Review form: Reviewer 2

Recommendation

Accept with minor revision (please list in comments)

Scientific importance: Is the manuscript an original and important contribution to its field?

Good

General interest: Is the paper of sufficient general interest?

Good

Quality of the paper: Is the overall quality of the paper suitable?

Good

Is the length of the paper justified?

Yes

Should the paper be seen by a specialist statistical reviewer?

No

Do you have any concerns about statistical analyses in this paper? If so, please specify them explicitly in your report.

Yes

It is a condition of publication that authors make their supporting data, code and materials available - either as supplementary material or hosted in an external repository. Please rate, if applicable, the supporting data on the following criteria.

Is it accessible?

Yes

Is it clear?

Yes

Is it adequate?

Yes

Do you have any ethical concerns with this paper?

No

Comments to the Author

Manuscript RSPB-2019-2646 entitled “Chemical signals act as the main reproductive barrier between sister and mimetic *Heliconius* butterflies” explores the role of wing phenotype and male chemical profile for mate choice and species recognition in a pair of coexisting mimetic species. The explorations of wing patterns find minor, non-significant differences between the species, but

more profound differences in the chemical profiles of the two species. Two behavioural experiments show that wing patterns do not alter female preference of males of the two species, but pheromones do, resulting in the authors suggesting that the pheromones function as the reproductive barrier between the two species.

This work is interesting, the experiments are well planned, and the manuscript is well written. Thus, it is indeed a good study, but considering a lot of the result are mostly corroborating existing knowledge, I am unsure about how well it fits to the scope of Proceedings B. The role of chemical signaling is well studied, and shown to be of great importance across a variety of insects (for example in CHC literature) and variety of situations, so finding it to be important in species where the colour patterns of the two species are largely identical, with similar studies having shown the same before (as cited in the paper), the results do not seem particularly novel. That aside, should the authors widen the perspective to discuss the role of chemical signaling in reproductive isolation on a larger scale, aside from *Heliconius* species, the manuscript might be a better fit to the wide readership of Proceedings B.

Major comments

1. The discussion is very focused on the specific cases of *Heliconius* with a few mentions of other Lepidoptera. But what do these findings mean for the evolution of mimetic species? How does it relate to sympatric speciation? How does using chemical cues for mate choice alter our understanding of mimicry, or for sexual selection? I believe the authors should make the discussion more explicit about the wider implications of these findings.
2. The authors could consider analyzing the behavioural studies using a GLMM with two fixed factors (treatment and species) and their interaction, to compare the responses of the two species directly (lines 213-222). This would reduce the number of models used for the data, and also provide a statistical test of whether the two species differ in their responses. Please also clearly state how the response variable was treated in the GLMM's (213-218). Was it used as a binary outcome or a proportion?
3. The reporting of statistics is lacking. First, across the manuscript, only p-values are reported. Please add information about the test statistics and the degrees of freedom (for example, lines 268-27). Secondly, there is no need to present the p-values with so many decimals. There are also inconsistencies in description of non-significant results: All colour pattern differences were not significant, yet the trends were extensively discussed (line 280-290), yet very little discussion is provided for the non-significant trends in the wing phenotype experiment (Line 297-302). Please be consistent with the treatment of non-significant trends.
4. No statistics are provided for tests of colour patterns (lines 288-290). Please ensure that all (even non-significant) effects are described with appropriate statistics.
5. The use of figures could be improved. Considering a lot of the results rely on information contained in Figure S7, I believe this figure should be presented in the manuscript. On the other hand, Figure 1A takes up a lot of space, but yet contains information that could easily be described in a short sentence in text without losing any accuracy. Considering the remarkable likeness of Figure 1B and Figure S7, where one contains significant differences and the other one does not, it would also be more transparent to show both in the manuscript itself. It also emphasizes the need for reporting of the parameter estimates and their standard errors, to understand why these very similar patterns seem to yield different results.
6. Line 327: please report the sample sizes per group.

Minor comments

1. Considering the general scope of the journal, I believe a short explanation for a few specific

terms would be beneficial. For example, perhaps at first mention of androconia (line 63), add a short description of it: I.e. "This is supported by the fact that males of *Heliconius* have species-specific mixtures in their male-limited scent scales (the androconia) on their wings." Similarly, the term "genital bouquet" could be explained (line 227).

2. The methods section could be restructured slightly for easier readability. I suggest moving the "Quantification of the wing phenotype" section (lines 119-149) to be presented prior to the information about larval rearing. To avoid confusion about which samples were used for which part of the study, I would also prefer the sentence on lines 97-99 to be presented within this section. The introduction to the hybrid crosses (lines 106-117) could also be given at the section where these individuals are used. Thus, keeping the information about the origin of the samples (i.e. moving the section "Wild sampling and interspecific crosses") in the same section as the study that uses them, following the methodology would be more straightforward.

3. Lines 207-211: It is a bit unclear to me what the purpose of the hybrids and the backcrosses are for this study. Please clarify the expectations and justifications of these treatments.

4. Is it necessary to introduce all the abbreviations described in lines 121-127? LM for landmark is used a total of 5 times in the document, and can lead to confusion with the standard abbreviation for linear models. While FW, HW and BC are used more often, I am not sure the introduction of these abbreviation does anything more than confuse the reader.

5. Line 42: replace the "the" with "a" in "...involve the combination..."

6. Line 60: Add "species" after *Heliconius*

7. Line 85-86: Instead of being unspecific, start with the exact questions that do need further investigation, and why it is important. Also, justify why we need to know points i and ii in lines 86-89 explicitly.

8. Line 145-146: Is it necessary to state that the differences were visualized with a box plot? Either show the plot or remove as unnecessary information.

9. Lines 221-222: To my understanding none of these packages allow for mixed effects models?

10. Line 234: Pheromone titres?

11. Lines 369- 390: Are these paragraphs necessary? Starting the discussion from results rather than summarizing the introduction in two paragraphs would focus the manuscript better.

12. Line 432: "a" should be "as"

Decision letter (RSPB-2019-2646.R0)

07-Jan-2020

Dear Ms González Rojas:

Your manuscript has now been peer reviewed and the reviews have been assessed by an Associate Editor. The reviewers' comments (not including confidential comments to the Editor) and the comments from the Associate Editor are included at the end of this email for your reference. As you will see, the reviewers and the Editors have raised some concerns with your manuscript and we would like to invite you to revise your manuscript to address them.

Research ethics:

Use of animals and field studies:

Please submit a copy of your revised paper within three weeks. If we do not hear from you within this time your manuscript will be rejected. If you are unable to meet this deadline please let us know as soon as possible, as we may be able to grant a short extension.

Best wishes,
Professor Gary Carvalho
mailto:proceedingsb@royalsociety.org

Associate Editor
Board Member: 1
Comments to Author:

Overall, both reviewers were positive about this work, but requested further clarifications in some aspects of the methods used and proposed a widening of the discussion for the wider readership of the journal. Please see reviewers comments for details.

Reviewer(s)' Comments to Author:

Referee: 1

Comments to the Author(s)

This manuscript investigated the importance of male wing colour and chemical composition of the androconia for female mating preference, and species discrimination, in two species of *Heliconius*.

It is clear that a lot of work has gone into this manuscript, and it has important implications for the study of species discrimination and sexual selection in mimicry rings, however I have some concerns about interpretation and terminology.

None of the trials where male chemical profile was manipulated resulted in matings, the authors therefore use acceptance and rejection behaviours to conclude that the chemical cues are responsible for female preference. This in and of itself is not an issue, perfuming experiments are tricky after all, but it is never mentioned in the discussion. Given that this is a key part of your study I think it warrants a little more discussion. In particular, as you found differences in genital gland extracts between the species but did not manipulate this in your behavioural experiments, perhaps that played a role? If different extracts play a role at different stages of courtship that may explain why no matings occurred despite females showing some acceptance behaviours .

I am also concerned with the language used to describe the “pheromone” composition results. If my reading is correct you have no way of knowing which components of the extracts are actually being detected and used in mating decisions by these females. Nevertheless they (by necessity) all have equal weight in your statistical analysis. This is a sensible approach to begin characterising the pheromones of these species, but you cannot then call this the pheromone bouquet, or conclude that the pheromone composition of the two species is significantly different – you can only say that the chemical composition of the extracts are different. Some of these chemicals (maybe even all) are pheromones, but you don’t yet know which.

Just to be clear- the androconial extract clearly contains pheromones, as blocking them prevented successful mating, but the PCA was performed on chemical composition. This means you have to be very careful when discussing your results as pheromone is the correct term to use for some aspects but not others.

There were also a few things in the methods that were unclear to me. When describing how you tested the longevity of the perfuming method on L200-205 you lost me entirely. Surely by dissecting and extracting the androconia in this way you also risk picking up the males own gland secretions? Can you give a little more information on why you used this method?

On L236-251 it is unclear if you used GC-FID, GC-MS or both.

L353 in the methods you say you used PCA and here you say discriminate analysis. To my knowledge they are similar but not the same. Please clarify.

I also have some more minor comments:

L31 I think you mean “preference of females for conspecific males”

L45 I suggest replacing “inform” with “can communicate”

L70 coexist

L74 “even so” suggests contrasting information. Replace with “and few hybrids etc etc”

L459 “inheritable” is a bit confusing, replace with “heritable”

Figures: I appreciate that you have limited figures but the decision not to have the figure for the behavioural response to chemical manipulation is a bit strange to me.

Referee: 2

Comments to the Author(s)

Manuscript RSPB-2019-2646 entitled “Chemical signals act as the main reproductive barrier between sister and mimetic *Heliconius* butterflies” explores the role of wing phenotype and male chemical profile for mate choice and species recognition in a pair of coexisting mimetic species. The explorations of wing patterns find minor, non-significant differences between the species, but more profound differences in the chemical profiles of the two species. Two behavioural experiments show that wing patterns do not alter female preference of males of the two species, but pheromones do, resulting in the authors suggesting that the pheromones function as the reproductive barrier between the two species.

This work is interesting, the experiments are well planned, and the manuscript is well written. Thus, it is indeed a good study, but considering a lot of the result are mostly corroborating existing knowledge, I am unsure about how well it fits to the scope of Proceedings B. The role of chemical signaling is well studied, and shown to be of great importance across a variety of insects (for example in CHC literature) and variety of situations, so finding it to be important in species where the colour patterns of the two species are largely identical, with similar studies having shown the same before (as cited in the paper), the results do not seem particularly novel. That aside, should the authors widen the perspective to discuss the role of chemical signaling in reproductive isolation on a larger scale, aside from *Heliconius* species, the manuscript might be a better fit to the wide readership of Proceedings B.

Major comments

1. The discussion is very focused on the specific cases of *Heliconius* with a few mentions of other Lepidoptera. But what do these findings mean for the evolution of mimetic species? How does it

relate to sympatric speciation? How does using chemical cues for mate choice alter our understanding of mimicry, or for sexual selection? I believe the authors should make the discussion more explicit about the wider implications of these findings.

2. The authors could consider analyzing the behavioural studies using a GLMM with two fixed factors (treatment and species) and their interaction, to compare the responses of the two species directly (lines 213-222). This would reduce the number of models used for the data, and also provide a statistical test of whether the two species differ in their responses. Please also clearly state how the response variable was treated in the GLMM's (213-218). Was it used as a binary outcome or a proportion?

3. The reporting of statistics is lacking. First, across the manuscript, only p-values are reported. Please add information about the test statistics and the degrees of freedom (for example, lines 268-27). Secondly, there is no need to present the p-values with so many decimals. There are also inconsistencies in description of non-significant results: All colour pattern differences were not significant, yet the trends were extensively discussed (line 280-290), yet very little discussion is provided for the non-significant trends in the wing phenotype experiment (Line 297-302). Please be consistent with the treatment of non-significant trends.

4. No statistics are provided for tests of colour patterns (lines 288-290). Please ensure that all (even non-significant) effects are described with appropriate statistics.

5. The use of figures could be improved. Considering a lot of the results rely on information contained in Figure S7, I believe this figure should be presented in the manuscript. On the other hand, Figure 1A takes up a lot of space, but yet contains information that could easily be described in a short sentence in text without losing any accuracy. Considering the remarkable likeness of Figure 1B and Figure S7, where one contains significant differences and the other one does not, it would also be more transparent to show both in the manuscript itself. It also emphasizes the need for reporting of the parameter estimates and their standard errors, to understand why these very similar patterns seem to yield different results.

6. Line 327: please report the sample sizes per group.

Minor comments

1. Considering the general scope of the journal, I believe a short explanation for a few specific terms would be beneficial. For example, perhaps at first mention of androconia (line 63), add a short description of it: i.e. "This is supported by the fact that males of *Heliconius* have species-specific mixtures in their male-limited scent scales (the androconia) on their wings." Similarly, the term "genital bouquet" could be explained (line 227).

2. The methods section could be restructured slightly for easier readability. I suggest moving the "Quantification of the wing phenotype" section (lines 119-149) to be presented prior to the information about larval rearing. To avoid confusion about which samples were used for which part of the study, I would also prefer the sentence on lines 97-99 to be presented within this section. The introduction to the hybrid crosses (lines 106-117) could also be given at the section where these individuals are used. Thus, keeping the information about the origin of the samples (i.e. moving the section "Wild sampling and interspecific crosses") in the same section as the study that uses them, following the methodology would be more straightforward.

3. Lines 207-211: It is a bit unclear to me what the purpose of the hybrids and the backcrosses are for this study. Please clarify the expectations and justifications of these treatments.

4. Is it necessary to introduce all the abbreviations described in lines 121-127? LM for landmark is used a total of 5 times in the document, and can lead to confusion with the standard abbreviation

for linear models. While FW, HW and BC are used more often, I am not sure the introduction of these abbreviation does anything more than confuse the reader.

5. Line 42: replace the “the” with “a” in “...involve the combination...”

6. Line 60: Add “species” after Heliconius

7. Line 85-86: Instead of being unspecific, start with the exact questions that do need further investigation, and why it is important. Also, justify why we need to know points i and ii in lines 86-89 explicitly.

8. Line 145-146: Is it necessary to state that the differences were visualized with a box plot? Either show the plot or remove as unnecessary information.

9. Lines 221-222: To my understanding none of these packages allow for mixed effects models?

10. Line 234: Pheromone titres?

11. Lines 369- 390: Are these paragraphs necessary? Starting the discussion from results rather than summarizing the introduction in two paragraphs would focus the manuscript better.

12. Line 432: “a” should be “as”

Author's Response to Decision Letter for (RSPB-2019-2646.R0)

See Appendix A.

RSPB-2019-2646.R1 (Revision)

Review form: Reviewer 1

Recommendation

Accept with minor revision (please list in comments)

Scientific importance: Is the manuscript an original and important contribution to its field?

Good

General interest: Is the paper of sufficient general interest?

Good

Quality of the paper: Is the overall quality of the paper suitable?

Acceptable

Is the length of the paper justified?

Yes

Should the paper be seen by a specialist statistical reviewer?

No

Do you have any concerns about statistical analyses in this paper? If so, please specify them explicitly in your report.

No

It is a condition of publication that authors make their supporting data, code and materials available - either as supplementary material or hosted in an external repository. Please rate, if applicable, the supporting data on the following criteria.

Is it accessible?

Yes

Is it clear?

Yes

Is it adequate?

Yes

Do you have any ethical concerns with this paper?

No

Comments to the Author

The authors have adequately addressed my previous concerns. I am however very alarmed that in responding to the (very good) advice of Reviewer 2, and their concern that these results are "mostly corroborating existing knowledge" they have tried to minimise and dismiss the existing research on chemical signalling in mate choice. They say in response to Reviewer 2 that "while many pheromones have been identified in moths, and some in butterflies, the actual role of these chemicals is not usually tested in behavioural experiments" which I could not disagree with more. To conclude that something is a pheromone you have to show behavioural responses! There is a whole industry based upon the behavioural responses of moths (and other insects) to sex pheromones - mating disruption.

This then comes up in their newly-added discussion L532-543. It is unclear if the authors are trying to say that work on the role of sex-pheromones in day-flying butterflies has been limited up until now, but the actual wording seems to suggest that the (ample) body of research on moths is still somehow scarce? I understand the urge to emphasise the novelty of the work but to me this is a baffling statement, directly contradicted by other sections of the discussion and the authors own reference section (I considered adding papers here to support my point but the authors have already provided many themselves, so I don't think it is necessary). I hope that it is simply a case of ambiguous wording but in any event it needs to be addressed and corrected.

I also have a few minor corrections as I think some typos slipped in during the review process: L86 now I know what you are trying to convey here I think you should say "a few hybrids" or "a low number of hybrids"

L100-101 "thus contributes to our understanding of"

L532-533 I think there is a mistake in here as it doesn't currently make sense

Decision letter (RSPB-2019-2646.R1)

06-Mar-2020

Dear Ms González Rojas:

I am writing to inform you that your manuscript # RSPB-2019-2646.R1 entitled "Chemical signals act as the main reproductive barrier between sister and mimetic *Heliconius* butterflies" has been rejected for publication in Proceedings B.

This action has been taken on the advice of referees, who have recommended that substantial revisions are necessary. With this in mind we would be happy to consider a resubmission, provided the comments of the referees are fully addressed. However please note that this is not a provisional acceptance.

Please find below the comments made by the referees, not including confidential reports to the Editor, which I hope you will find useful. Please note that it is very unusual, for us to invite a 2nd complete round of resubmission. The decision has been based on the potential of your manuscript, though there is a strong need for a thorough consideration of comments from the referee. Please note, that while we provide the invitation for your consideration, as ever, it does not guarantee eventual acceptance. While we will do our best to appoint the original reviewer, this cannot be guaranteed, and on this occasion we will need to invite one further new reviewer. I appreciate that such action is extending our usual peer review process, and of course you must decide whether you are happy to proceed.

If you do choose to resubmit your manuscript, please upload the following:

Sincerely,
 Professor Gary Carvalho
 Editor, Proceedings B
proceedingsb@royalsociety.org

Reviewer(s)' Comments to Author:

Referee: 1

Comments to the Author(s)

The authors have adequately addressed my previous concerns. I am however very alarmed that in responding to the (very good) advice of Reviewer 2, and their concern that these results are "mostly corroborating existing knowledge" they have tried to minimise and dismiss the existing research on chemical signalling in mate choice. They say in response to Reviewer 2 that "while many pheromones have been identified in moths, and some in butterflies, the actual role of these chemicals is not usually tested in behavioural experiments" which I could not disagree with more. To conclude that something is a pheromone you have to show behavioural responses! There is a whole industry based upon the behavioural responses of moths (and other insects) to sex pheromones - mating disruption.

This then comes up in their newly-added discussion L532-543. It is unclear if the authors are trying to say that work on the role of sex-pheromones in day-flying butterflies has been limited

up until now, but the actual wording seems to suggest that the (ample) body of research on moths is still somehow scarce? I understand the urge to emphasise the novelty of the work but to me this is a baffling statement, directly contradicted by other sections of the discussion and the authors own reference section (I considered adding papers here to support my point but the authors have already provided many themselves, so I don't think it is necessary). I hope that it is simply a case of ambiguous wording but in any event it needs to be addressed and corrected.

I also have a few minor corrections as I think some typos slipped in during the review process:
 L86 now I know what you are trying to convey here I think you should say "a few hybrids" or "a low number of hybrids"

L100-101 "thus contributes to our understanding of"

L532-533 I think there is a mistake in here as it doesn't currently make sense

Author's Response to Decision Letter for (RSPB-2019-2646.R1)

See Appendix B.

RSPB-2020-0587.R0

Review form: Reviewer 1

Recommendation

Accept as is

Scientific importance: Is the manuscript an original and important contribution to its field?

Good

General interest: Is the paper of sufficient general interest?

Good

Quality of the paper: Is the overall quality of the paper suitable?

Good

Is the length of the paper justified?

Yes

Should the paper be seen by a specialist statistical reviewer?

No

Do you have any concerns about statistical analyses in this paper? If so, please specify them explicitly in your report.

No

It is a condition of publication that authors make their supporting data, code and materials available - either as supplementary material or hosted in an external repository. Please rate, if applicable, the supporting data on the following criteria.

Is it accessible?

Yes

Is it clear?

Yes

Is it adequate?

Yes

Do you have any ethical concerns with this paper?

No

Comments to the Author

I'm glad to see that the authors took my previous comments seriously. The changes they made have fully clarified the position of this study in the wider field of insect pheromone research, while still emphasising its novelty in the context of day-flying butterflies. I am happy to recommend the manuscript for publication in its current form.

Review form: Reviewer 2

Recommendation

Accept with minor revision (please list in comments)

Scientific importance: Is the manuscript an original and important contribution to its field?

Good

General interest: Is the paper of sufficient general interest?

Good

Quality of the paper: Is the overall quality of the paper suitable?

Excellent

Is the length of the paper justified?

Yes

Should the paper be seen by a specialist statistical reviewer?

No

Do you have any concerns about statistical analyses in this paper? If so, please specify them explicitly in your report.

No

It is a condition of publication that authors make their supporting data, code and materials available - either as supplementary material or hosted in an external repository. Please rate, if applicable, the supporting data on the following criteria.

Is it accessible?

Yes

Is it clear?

Yes

Is it adequate?

Yes

Do you have any ethical concerns with this paper?

No

Comments to the Author

The authors of manuscript RSPB-2020-0587 have addressed my previous concerns as well as those of Reviewer 1 sufficiently. The current version of the manuscript has a wider perspective, and I believe the corrections make the case for the importance of this work much clearer for studies of reproductive isolation beyond *Heliconius* butterflies. I also believe the use of graphs and the reporting of statistics is now appropriate and more transparent. I have a few minor comments that could improve the clarity in a few places, but otherwise this manuscript is, in my opinion, is ready for publication.

Minor comments

1. Line 107-108: What kind of variables is the PCA performed on?
2. Line 119-120: I still believe mentioning that the differences were visualized with a boxplot adds no necessary information to the manuscript.
3. Line 291: The F-values and df still missing from this particular result.
4. Line 299: move the sample sizes to after "perfumed males" for clarity.
5. Line 318: Consider adding "in most of the traits measured" at the end of this sentence.
6. Line 365-378: These paragraphs could be combined into one to keep the background information to one paragraph at the beginning of the Discussion.
7. Line 435: Unclear wording, as it could be read as "better" implying something about the quality of the male. Perhaps this should be: "Similarly, hybrid females (F1 and backcrosses) accepted males of *H. t. florenciae* more readily than those of *H. m. malleti*..." ?
8. Line 447-449: Please make it clear that this is a result of a previous study, not something that was quantified in this paper.

Decision letter (RSPB-2020-0587.R0)

03-Apr-2020

Dear Ms González Rojas

I am pleased to inform you that your manuscript RSPB-2020-0587 entitled "Chemical signals act as the main reproductive barrier between sister and mimetic *Heliconius* butterflies" has been accepted for publication in Proceedings B.

The referee(s) have recommended publication, but also suggest some minor revisions to your manuscript. Therefore, I invite you to respond to the referee(s)' comments and revise your manuscript. Because the schedule for publication is very tight, it is a condition of publication that you submit the revised version of your manuscript within 7 days. If you do not think you will be able to meet this date please let us know.

To revise your manuscript, log into <https://mc.manuscriptcentral.com/prsb> and enter your Author Centre, where you will find your manuscript title listed under "Manuscripts with

Decisions." Under "Actions," click on "Create a Revision." Your manuscript number has been appended to denote a revision. You will be unable to make your revisions on the originally submitted version of the manuscript. Instead, revise your manuscript and upload a new version through your Author Centre.

If you wish to submit your data to Dryad (<http://datadryad.org/>) and have not already done so

you can submit your data via this link

[http://datadryad.org/submit?journalID=RSPB&manu=\(Document not available\)](http://datadryad.org/submit?journalID=RSPB&manu=(Document+not+available)) which will take you to your unique entry in the Dryad repository. If you have already submitted your data to dryad you can make any necessary revisions to your dataset by following the above link. Please see <https://royalsociety.org/journals/ethics-policies/data-sharing-mining/> for more details.

Sincerely,
Professor Gary Carvalho
mailto: proceedingsb@royalsociety.org

Reviewer(s)' Comments to Author:

Referee: 2

Comments to the Author(s).

The authors of manuscript RSPB-2020-0587 have addressed my previous concerns as well as those of Reviewer 1 sufficiently. The current version of the manuscript has a wider perspective, and I believe the corrections make the case for the importance of this work much clearer for studies of reproductive isolation beyond *Heliconius* butterflies. I also believe the use of graphs and the reporting of statistics is now appropriate and more transparent. I have a few minor comments that could improve the clarity in a few places, but otherwise this manuscript is, in my opinion, is ready for publication.

Minor comments

1. Line 107-108: What kind of variables is the PCA performed on?
2. Line 119-120: I still believe mentioning that the differences were visualized with a boxplot adds no necessary information to the manuscript.
3. Line 291: The F-values and df still missing from this particular result.
4. Line 299: move the sample sizes to after “perfumed males” for clarity.
5. Line 318: Consider adding “in most of the traits measured” at the end of this sentence.
6. Line 365-378: These paragraphs could be combined into one to keep the background information to one paragraph at the beginning of the Discussion.
7. Line 435: Unclear wording, as it could be read as “better” implying something about the quality of the male. Perhaps this should be: “Similarly, hybrid females (F1 and backcrosses) accepted males of *H. t. florenciae* more readily than those of *H. m. malleti*...” ?
8. Line 447-449: Please make it clear that this is a result of a previous study, not something that was quantified in this paper.

Referee: 1

Comments to the Author(s).

I'm glad to see that the authors took my previous comments seriously. The changes they made have fully clarified the position of this study in the wider field of insect pheromone research, while still emphasising its novelty in the context of day-flying butterflies. I am happy to recommend the manuscript for publication in its current form.

Author's Response to Decision Letter for (RSPB-2020-0587.R0)

See Appendix C.

Decision letter (RSPB-2020-0587.R1)

09-Apr-2020

Dear Ms González Rojas

I am pleased to inform you that your manuscript entitled "Chemical signals act as the main reproductive barrier between sister and mimetic *Heliconius* butterflies" has been accepted for publication in Proceedings B.

Open Access

You are invited to opt for Open Access, making your freely available to all as soon as it is ready for publication under a CCBY licence. Our article processing charge for Open Access is £1700. Corresponding authors from member institutions (<http://royalsocietypublishing.org/site/librarians/allmembers.xhtml>) receive a 25% discount to these charges. For more information please visit <http://royalsocietypublishing.org/open-access>.

Paper charges

All supplementary materials accompanying an accepted article will be treated as in their final form. They will be published alongside the paper on the journal website and posted on the online

figshare repository. Files on figshare will be made available approximately one week before the accompanying article so that the supplementary material can be attributed a unique DOI.

Sincerely,
Proceedings B
<mailto:proceedingsb@royalsociety.org>

Appendix A

Professor Gary Carvalho
Editor
Proceedings of The Royal Society B

Subject: Revision and resubmission of manuscript ID RSPB-2019-2646

Dear Professor Gary Carvalho,

We are glad to submit a revised version of our manuscript entitled “Chemical signals act as the main reproductive barrier between sister and mimetic *Heliconius* butterflies”.

We are very grateful to the reviewers for their positive and helpful comments. Following their advice and your suggestion, we have reworded some sections, added extra information and/or discussion and reorganized figures to improve the readability and facilitate interpretation. Please see our point-by-point response below.

We believe this leads to a significantly improved manuscript and hope that the changes will allow a clearer understanding of our study.

We thank you and the reviewers for your time and consideration and look forward to hearing from you soon.

Sincerely,

Gonzalez-Rojas et al.

COMMENTS FROM REVIEWER 1:

Comments to the Author(s):

*This manuscript investigated the importance of male wing colour and chemical composition of the androconia for female mating preference, and species discrimination, in two species of *Heliconius*. It is clear that a lot of work has gone into this manuscript, and it has important implications for the study of species discrimination and sexual selection in mimicry rings, however I have some concerns about interpretation and terminology.*

We would like to thank the reviewer for recognizing the effort and implications of our study.

- *None of the trials where male chemical profile was manipulated resulted in matings, the authors therefore use acceptance and rejection behaviours to conclude that the chemical cues are responsible for female preference. This in and of itself is not an issue, perfuming experiments are tricky after all, but it is never mentioned in the discussion. Given that this is a key part of your study I think it warrants a little more discussion. In particular, as you found differences in genital gland extracts between the species but did not manipulate this in your behavioural experiments, perhaps that played a role? If different extracts play a role at different stages of courtship that may explain why no matings occurred despite females showing some acceptance behaviours.”*

While there are reports on the role of the abdominal gland in mate recognition, our experimental design does not allow us to test for such role. In our design, a female was tested against two males from her own species. Both males were blocked with nail varnish and then both were re-perfumed, one with the heterospecific androconial scent and the second with the conspecific androconial scent. Because both males were from the same species, they didn't differ in the abdominal gland scent (which was never altered or blocked) and therefore, we could not test whether the chemicals of the abdominal gland play a role in mate discrimination. Despite this, our results showed that the presence of the abdominal scent alone is not enough for mating, and the presence of the wing androconial scent is always necessary for mating to happen. This is now discussed in lines 406-420.

- *I am also concerned with the language used to describe the “pheromone” composition results. If my reading is correct you have no way of knowing which components of the extracts are actually being detected and used in mating decisions by these females. Nevertheless they (by necessity) all have equal weight in your statistical analysis. This is a sensible approach to begin characterising the pheromones of these species, but you cannot then call this the pheromone bouquet, or conclude that the pheromone composition of the two species is significantly different – you can only say that the chemical composition of the extracts are different. Some of these chemicals (maybe even all) are pheromones, but you don't yet know which. Just to be clear- the androconial extract clearly contains pheromones, as blocking them prevented successful mating, but the PCA was performed on chemical composition. This means you have to be very careful when discussing your results as pheromone is the correct term to use for some aspects but not others.*

We replaced the word pheromone along the manuscript where pertinent.

- *There were also a few things in the methods that were unclear to me. When describing how you tested the longevity of the perfuming method on L200-205 you lost me entirely. Surely by dissecting and extracting the androconia in this way you also risk picking up the males own gland secretions? Can you give a little more information on why you used this method?”*

Our previous experiments (Darragh, et al. 2017, PeerJ) and those of Costanzo & Monteiro (2007, Proc Biol Sci) show that applying nail varnish on the wing androconia effectively blocks the emission of chemicals from the androconia. As such, when testing the longevity of the perfume, the GC/MS measures should correspond to the androconial perfume applied to the male (and not the ones produced by the androconia of such male, since they were blocked beforehand). In any case, these experiments sought to test how long the applied perfume lasted for, and in fact, they revealed that the perfume evaporates in a short period of time (i.e. the concentration of the compounds decreases, Table S4 and Figure S7).

- *On L236-251 it is unclear if you used GC-FID, GC-MS or both.*

Thanks for bringing this to our attention. We used only GC-MS, and we made the pertinent corrections in the methods section. See lines 243-252.

- *L353 in the methods you say you used PCA and here you say discriminate analysis. To my knowledge they are similar but not the same. Please clarify.*

We reworded the methods section (see lines 260-263) to make our analysis clearer. Briefly, we conducted a PCA and then a discriminant analysis on those components that explained most of the variance.

Minor comments

- *“L31 I think you mean “preference of females for conspecific males””*

Done

- *“L45 I suggest replacing “inform” with “can communicate””*

Done

- *“L70 coexist”*

Done

- *“L74 “even so” suggests contrasting information. Replace with “and few hybrids etc etc””*

Thanks for the suggestion but we actually want to make a contrast: although these are ‘good species’, some hybrids are found (in low frequency).

- *“L459 “inheritable” is a bit confusing, replace with “heritable””*

Done

- *Figures: I appreciate that you have limited figures but the decision not to have the figure for the behavioural response to chemical manipulation is a bit strange to me.*

Following your suggestion, we moved the figure to main text (Figure 1B).

COMMENTS FROM REVIEWER 2

Comments to the Author(s):

Manuscript RSPB-2019-2646 entitled “Chemical signals act as the main reproductive barrier between sister and mimetic Heliconius butterflies” explores the role of wing phenotype and male chemical profile for mate choice and species recognition in a pair of coexisting mimetic species. The explorations of wing patterns find minor, non-significant differences between the species, but more profound differences in the chemical profiles of the two species. Two behavioural experiments show that wing patterns do not alter female preference of males of the two species, but pheromones do, resulting in the authors suggesting that the pheromones function as the reproductive barrier between the two species.

This work is interesting, the experiments are well planned, and the manuscript is well written. Thus, it is indeed a good study, but considering a lot of the result are mostly corroborating existing knowledge, I am unsure about how well it fits to the scope of Proceedings B. The role of chemical signaling is well studied, and shown to be of great importance across a variety of insects (for example in CHC literature) and variety of situations, so finding it to be important in species where the colour patterns of the two species are largely identical, with similar studies having shown the same before (as cited in the paper), the results do not seem particularly novel. That aside, should the authors widen the perspective to discuss the role of chemical signaling in reproductive isolation on a larger scale, aside from Heliconius species, the manuscript might be a better fit to the wide readership of Proceedings B. The discussion is very focused on the specific cases of Heliconius with a few mentions of other Lepidoptera. But what do these findings mean for the evolution of mimetic species? How does it relate to sympatric speciation? How does using chemical cues for mate choice alter our understanding of mimicry, or for sexual selection? I believe the authors should make the discussion more explicit about the wider implications of these findings.”

Thanks for the reviewer for his/her valuable insight and comments since they have improved the paper significantly. The discussion was broadened as suggested, trying to include information of taxa other than *Heliconius* and expanding the scope by discussing implications of our results in speciation. We think the study is novel as it provides new information that helps understand the importance of chemical signals in reproductive isolation and speciation, especially in cases where other isolating barriers (such as colour pattern) are secondary lost. Furthermore, this is one of the first studies that tests female preference for male scents and suggests that this preference may have a simple genetic basis. In fact, sexual chemical production (by males) also seems to have a simple inheritance, and if both of these traits (production and preference) happen to be physically

linked in the genome, then reproductive isolation would easily be achieved and maintained in the presence of gene flow (although this remains to be tested in future studies). Finally, our study is one of the few that experimentally tests the role of chemical scents in mate choice discrimination in Lepidoptera (while many pheromones have been identified in moths, and some in butterflies, the actual role of these chemicals is not usually tested in behavioral experiments). In response to this comment, we have extended the discussion by introducing lines 373-392, 406-420, 463-478, 487-492.

- *The authors could consider analyzing the behavioural studies using a GLMM with two fixed factors (treatment and species) and their interaction, to compare the responses of the two species directly (lines 213-222). This would reduce the number of models used for the data, and also provide a statistical test of whether the two species differ in their responses. Please also clearly state how the response variable was treated in the GLMM's (213-218). Was it used as a binary outcome or a proportion?"*

Thanks for suggesting a complex model that tests two fixed factors and their interaction. However, the experimental design does not allow to directly test the variable 'species'. As we mentioned in the methods, each behavioural experiment was performed independently per species (i.e. a single female was tested with two males from her own species – treatment and control). This design makes it unnecessary to test the effect of the variable 'species' or its interaction with the variable 'treatment'.

The response variable was treated as a binary outcome using family=binomial in the model (as previously done in Darragh, et al. 2017, PeerJ).

- *The reporting of statistics is lacking. First, across the manuscript, only p-values are reported. Please add information about the test statistics and the degrees of freedom (for example, lines 268-27). Secondly, there is no need to present the p-values with so many decimals.*

We thank the reviewer for having drawn our attention on this. We included test statistics across the manuscript, including degrees of freedom. Also, we now report all tests in Table S2, S3 and S5. As suggested, p-values are now reported as higher or lower than the significance cut-off (0.01).

- *There are also inconsistencies in description of non-significant results: All colour pattern differences were not significant, yet the trends were extensively discussed (line 280-290), yet very little discussion is provided for the non-significant trends in the wing phenotype experiment (Line 297-302). Please be consistent with the treatment of non-significant trends.*

The reviewer is right. Although in the 'colour pattern' section we wanted to emphasize the existence of subtle differences (that may be interesting for some readers), we decided to remove detailed explanations of these since colour pattern as a whole does not differ

between species. By introducing this change, we keep consistency in the treatment of the non-significant trends.

- *“No statistics are provided for tests of colour patterns (lines 288-290). Please ensure that all (even non-significant) effects are described with appropriate statistics.”*

Thank you for this comment. These values were included to the figure legend (Figure S5) as requested.

- *The use of figures could be improved. Considering a lot of the results rely on information contained in Figure S7, I believe this figure should be presented in the manuscript. On the other hand, Figure 1A takes up a lot of space, but yet contains information that could easily be described in a short sentence in text without losing any accuracy. Considering the remarkable likeness of Figure 1B and Figure S7, where one contains significant differences and the other one does not, it would also be more transparent to show both in the manuscript itself. It also emphasizes the need for reporting of the parameter estimates and their standard errors, to understand why these very similar patterns seem to yield different results.*

Change done as suggested.

- *Line 327: please report the sample sizes per group.*

Numbers added as requested. See lines 331-332.

Minor comments

- *Considering the general scope of the journal, I believe a short explanation for a few specific terms would be beneficial. For example, perhaps at first mention of androconia (line 63), add a short description of it: I.e. “This is supported by the fact that males of Heliconius have species-specific mixtures in their male-limited scent scales (the androconia) on their wings.” Similarly, the term “genital bouquet” could be explained (line 227).*

We have added this information as requested by the reviewer lines 63 and 233-234.

- *The methods section could be restructured slightly for easier readability. I suggest moving the “Quantification of the wing phenotype” section (lines 119-149) to be presented prior to the information about larval rearing. To avoid confusion about which samples were used for which part of the study, I would also prefer the sentence on lines 97-99 to be presented within this section. The introduction to the hybrid crosses (lines 106-117) could also be given at the section where these individuals are used. Thus, keeping the information about the origin of the samples (i.e. moving the section “Wild*

sampling and interspecific crosses”) in the same section as the study that uses them, following the methodology would be more straightforward.

Following the suggestion, “Quantification of the wing phenotype” is now at the beginning of the methods’ section (lines 95-126) and text previously on lines 97-99 was moved (now in lines 97-99). However, we prefer to keep the description of wild individuals and crosses in a single section to avoid redundancy in further sections.

- *Lines 207-211: It is a bit unclear to me what the purpose of the hybrids and the backcrosses are for this study. Please clarify the expectations and justifications of these treatments.*

These crosses aim to show the inheritance pattern of both, the chemical production and the preference for chemicals. In particular, this is the first study in *Heliconius* that shows some data on the inheritance pattern of the female preference for chemicals. As we mention in the discussion, from these crosses we could infer a simple genetic bases for both of these traits. We now highlight the importance of these results in the context of speciation (see lines 444-478, 484-492)

- *Is it necessary to introduce all the abbreviations described in lines 121-127? LM for landmark is used a total of 5 times in the document, and can lead to confusion with the standard abbreviation for linear models. While FW, HW and BC are used more often, I am not sure the introduction of these abbreviation does anything more than confuse the reader.*

Thank you for this suggestion. The text was changed accordingly.

- *Line 42: replace the “the” with “a” in “...involve the combination...”*

Done

- *Line 60: Add “species” after Heliconius”*

Done

- *Line 85-86: Instead of being unspecific, start with the exact questions that do need further investigation, and why it is important. Also, justify why we need to know points *i* and *ii* in lines 86-89 explicitly.*

Change done as suggested. See lines 85-89.

- *Line 145-146: Is it necessary to state that the differences were visualized with a box plot? Either show the plot or remove as unnecessary information.*

This information was removed.

- *Lines 221-222: To my understanding none of these packages allow for mixed effects models?*

The reviewer is right. We used the “lme4” package, which allow mixed models with the “glmer” function. We now mention this package in line 228.

- *Line 234: Pheromone titres?*

The text has been changed accordingly.

- *Lines 369- 390: Are these paragraphs necessary? Starting the discussion from results rather than summarizing the introduction in two paragraphs would focus the manuscript better.*

Paragraphs removed as suggested. Discussion changed.

- *Line 432: “a” should be “as”*

Done

Appendix B

Professor Gary Carvalho
Editor
Proceedings of The Royal Society B

Subject: Revision and resubmission of manuscript ID RSPB-2020-0587

Dear Professor Gary Carvalho,

We are glad to re-submit a revised version of our manuscript entitled "Chemical signals act as the main reproductive barrier between sister and mimetic *Heliconius* butterflies" - ID RSPB-2019-2646.R1.

We would like to thank you for the unusual opportunity you gave us of re-submit our manuscript, and the reviewer for being so helpful. Following his/her advice we have reworded the discussion, particularly the section that created confusion. Please see our point-by-point response below.

We hope you find this new version suitable for publication

Sincerely,

Gonzalez-Rojas et al.

COMMENTS FROM REVIEWER 1:

Reviewer(s)' Comments to Author:

The authors have adequately addressed my previous concerns. I am however very alarmed that in responding to the (very good) advice of Reviewer 2, and their concern that these results are "mostly corroborating existing knowledge" they have tried to minimise and dismiss the existing research on chemical signalling in mate choice. They say in response to Reviewer 2 that "while many pheromones have been identified in moths, and some in butterflies, the actual role of these chemicals is not usually tested in behavioural experiments" which I could not disagree with more. To conclude that something is a pheromone you have to show behavioural responses! There is a whole industry based upon the behavioural responses of moths (and other insects) to sex pheromones - mating disruption.

This then comes up in their newly-added discussion L532-543. It is unclear if the authors are trying to say that work on the role of sex-pheromones in day-flying butterflies has been limited up until now, but the actual wording seems to suggest

that the (ample) body of research on moths is still somehow scarce? I understand the urge to emphasise the novelty of the work but to me this is a baffling statement, directly contradicted by other sections of the discussion and the authors own reference section (I considered adding papers here to support my point but the authors have already provided many themselves, so I don't think it is necessary). I hope that it is simply a case of ambiguous wording but in any event it needs to be addressed and corrected.

We are sorry for the misunderstanding. We did not mean to dismiss or minimise the work previously done. We reworded the section to reflect that, although the evidence from moths is ample, more studies are needed in butterflies.

I also have a few minor corrections as I think some typos slipped in during the review process:

- L86 now I know what you are trying to convey here I think you should say "a few hybrids" or "a low number of hybrids"

Change done as suggested.

L100-101 "thus contributes to our understanding of"

The text was changed accordingly.

- L532-533 I think there is a mistake in here as it doesn't currently make sense

Thanks for bringing this to our attention. The text was changed.

Appendix C

Professor Gary Carvalho
Editor
Proceedings of The Royal Society B

Subject: Revision of manuscript accepted ID RSPB-2020-0587

Dear Professor Gary Carvalho,

Thanks for your response and for the good news. We corrected the minor points raised by the reviewer two, and you can find our response below.

Once again thank you very much for your help in this process

Sincerely,

González-Rojas et al.

COMMENTS FROM REVIEWER 1:

Reviewer(s)' Comments to Author:

I'm glad to see that the authors took my previous comments seriously. The changes they made have fully clarified the position of this study in the wider field of insect pheromone research, while still emphasising its novelty in the context of day-flying butterflies. I am happy to recommend the manuscript for publication in its current form.

We would like to thank you for your positive and helpful comments that improved our manuscript.

COMMENTS FROM REVIEWER 2:

Reviewer(s)' Comments to Author:

The authors of manuscript RSPB-2020-0587 have addressed my previous concerns as well as those of Reviewer 1 sufficiently. The current version of the manuscript has a wider perspective, and I believe the corrections make the case for the importance of this work much clearer for studies of reproductive isolation beyond *Heliconius* butterflies. I also believe the use of graphs and the reporting of statistics is now appropriate and more transparent. I have a few minor comments that could improve the clarity in a few places, but otherwise this manuscript is, in my opinion, is ready for publication.

Thanks for your valuable and insightful comments, which greatly improved our manuscript.

Minor comments

1. Line 107-108: What kind of variables is the PCA performed on?

We added the information required by the reviewer. Patternize extracts the total wing area in which the pattern is expressed, then the PCA is performed on the binary representation of the aligned colour pattern rasters obtained from each sample. See lines 108-109.

2. Line 119-120: I still believe mentioning that the differences were visualized with a boxplot adds no necessary information to the manuscript.

This information was removed.

3. Line 291: The F-values and df still missing from this particular result.

This is the result of the binomial test and not the GLMM. We clarified this by saying “exact binomial test $p=0.55$ ” in line 293.

4. Line 299: move the sample sizes to after “perfumed males” for clarity.

Change done accordingly (see line 301).

5. Line 318: Consider adding “in most of the traits measured” at the end of this sentence.

Information added (see line 320).

6. Line 365-378: These paragraphs could be combined into one to keep the background information to one paragraph at the beginning of the Discussion.

Paragraphs combined in a single one as suggested.

7. Line 435: Unclear wording, as it could be read as “better” implying something about the quality of the male. Perhaps this should be: “Similarly, hybrid females (F1 and backcrosses) accepted males of *H. t. florenzia* more readily than those of *H. m. malleti*...” ?

Change done as suggested (see line 437).

8. Line 447-449: Please make it clear that this is a result of a previous study, not something that was quantified in this paper.

Word 'previous' added for clarity (see line 449).